# De-Sealing Reverses Habitat Decay More Than Increasing Groundcover Vegetation

Virginia Thompson Couch [1,*], Stefano Salata [2], Nicel Saygin [3], Anne Frary [4] and Bertan Arslan [3]

1 Department of Architecture, Faculty of Architecture, Izmir Institute of Technology, Izmir 35430, Türkiye
2 Department of Architecture and Urban Studies (DAStU), Lab PPTE, Politecnico di Milano, 20133 Milano, Italy; stefano.salata@polimi.it
3 Department of City and Regional Planning, Faculty of Architecture, Izmir Institute of Technology, Izmir 35430, Türkiye; nicelsaygin@iyte.edu.tr (N.S.); bertanaslan@iyte.edu.tr (B.A.)
4 Department of Molecular Biology and Genetics, Faculty of Science, Izmir Institute of Technology, Izmir 35430, Türkiye; annefrary@iyte.edu.tr
* Correspondence: virginiacouch@iyte.edu.tr

**Abstract:** Modeling ecosystem services is a growing trend in scientific research, and Nature-based Solutions (NbSs) are increasingly used by land-use planners and environmental designers to achieve improved adaptation to climate change and mitigation of the negative effects of climate change. Predictions of ecological benefits of NbSs are needed early in design to support decision making. In this study, we used ecological analysis to predict the benefits of two NbSs applied to a university masterplan and adjusted our preliminary design strategy according to the first modeling results. Our Area of Interest was the IZTECH campus, which is located in a rural area of the eastern Mediterranean region (Izmir/Turkey). A primary design goal was to improve habitat quality by revitalizing soil. Customized analysis of the Baseline Condition and two NbSs scenarios was achieved by using local values obtained from a high-resolution photogrammetric scan of the catchment to produce flow accumulation and habitat quality indexes. Results indicate that anthropogenic features are the primary cause of habitat decay and that decreasing imperviousness reduces habitat decay significantly more than adding vegetation. This study creates a method of supporting sustainability goals by quickly testing alternative NbSs. The main innovation is demonstrating that early approximation of the ecological benefits of NbSs can inform preliminary design strategy. The proposed model may be calibrated to address specific environmental challenges of a given location and test other forms of NbSs.

**Keywords:** sustainability; predictive modeling; nature-based solutions; GIS-based ecological analysis; habitat quality; habitat decay; anthropogenic footprint; de-sealing

## 1. Introduction

Global warming is causing threats such as less frequent rain events of longer duration and an increase in the frequency and size of wildfires in certain ecosystems [1–3]. Precisely targeted strategies are needed to adapt to these threats [4,5]. Future land-use-planning policies must be based on environmental health, informed by customized ecological analysis of smaller areas, and easy to apply [6,7]. Sustainability assessment indicators must be integrated with the spatial dimensions of specific areas to assess sustainability more effectively [8]. GIS-based ecological analysis allows the assessment of environmental changes within small areas over short periods [8,9]. Using GIS-based analysis, it is now possible to make land-use decisions based on predictive modeling of specific locations and test different solutions to determine which provide the greatest ecological benefits [10].

Universities are increasingly aware of their potential roles in sustainability efforts, such as reducing harm to the environment by implementing sustainable practices, producing knowledge, and developing new technologies [11]. University campuses often cover wide

areas and therefore have potentially large environmental impacts. Recent studies indicate that habitat decay is a driver of general environmental conditions because it has a direct relation with the well-being of citizens [12–14]. This study uses spatial evaluation of a habitat quality model [15] as a Decision-Making Support System to guide preliminary design during the development of a Nature-based Solutions (NbSs) masterplan for a university campus located in a semi-arid rural area. The habitat quality model is an ecosystem service spatial model that we employed as a proxy of biodiversity, because it considers the impact of anthropogenic factors on the natural landscape [16,17]. The study was conducted as part of the "IZTECH Living Laboratory and Ecological Park" interdisciplinary research project.

### 1.1. Area of Interest: IZTECH Campus, Cesme-Karaburun Peninsula, Turkey

The Eastern Mediterranean region is highly vulnerable to climate change and experiences effects such as increased drought, flash flooding, and loss of biodiversity [18–20]. The region was deforested centuries ago for building and agriculture [21]. With climate change, the summer temperatures in this region are increasing, and there are long periods of drought coupled with less frequent but more intense rain events [22]. According to a 2016 study conducted by the Basque Center for Climate Change, Istanbul is Europe's most vulnerable coastal city and Izmir ranks third in terms of vulnerability to climate change [23]. The average temperature of the Izmir Province for many years (1938–2018) was 17.9 °C, and annual average temperatures demonstrate a trend of increasing 1.7 °C/100 years [24]. The highest maximum temperature in Izmir demonstrates a trend of increasing by 1.2 °C/100 years, and an increasing trend has also been observed in the number of tropical nights in Izmir [24]. Tropical nights are defined as days when the night-time temperature does not fall below 20 °C. A high number of tropical nights is related to poor health outcomes [25].

Our Area of Interest (AOI) is a university campus on the Cesme-Karaburun Peninsula in the Metropolitan Area of Izmir, Turkey, in the eastern Aegean region (Figure 1). The terrain of the peninsula is mountainous, and the soil is predominantly limey [26]. Centuries of deforestation, fires, and erosion have left a thin layer of topsoil covered with evergreen sclerophyll bushes and shrubs (maquis) growing in low phrygana formations. This type of vegetation is characterized by a scrubby growth habit and small, leathery leaves. Posidonia seaweed (*Posidonia oceanica*) is found in the waters of the Gulbahce Bay. Fauna that can be seen today in the AOI include wild boar, foxes, porcupines, tortoises, chameleons, snakes, lizards, scorpions, hawks, owls, partridges, wagtails, and European kestrels. The population of the peninsula is increasing rapidly, and in summer months, the demand for freshwater exceeds the available supply [27]. In the winter of 2021, the population of the coastal town of Cesme, at the western tip of the peninsula, was 48,167; in the summer of 2021, the population was approximately 20 times greater [28]. The town of Cesme has begun taking water from the wells of nearby agricultural villages such as Ildırı to meet the increasing demand for drinking water, and there are plans to begin desalinating seawater for Cesme within a few years [29]. According to the Izmir Province Disaster Risk Reduction Plan, extreme heat, drought, and forest fires are among the highest climate risks in Izmir [24]. Drought-stressed vegetation is more vulnerable to wildfires and less able to sustain its chemical defenses against insect infestation [30]. In August 2019, 500 ha of land in the rural areas of the Izmir region was damaged as a result of fires [31].

In addition to the negative impacts of climate change, the ecosystem of the peninsula is threatened by human activities, such as a surge in construction [32], mining activities [33], and the location of fish farms too close together and too close to shore [34]. There is an immediate need for economic and planning policies based on an environmentally sound perspective [35], one that includes protection of freshwater resources, soil revitalization, sustainable agriculture and fishing, restrictions on construction, and the creation of large nature reserves. The peninsula suffers from neoliberal policies of environmental governance that gravitate towards more commodification and less conservation of natural areas [36]. Some of the recent interruptions of natural areas are caused by the privatization of the

wind energy infrastructure, which is a result of deregulation and re-regulation of the energy market in Turkey [36]. Laws that afforded environmental and natural protection in the past have been revised to expand the use of natural areas and natural resources by private companies [36]. In the past few decades, significant construction has occurred in environmental protection areas such as freshwater stream beds, wetlands, coastal areas, forests, pastures, olive (*Olea europaea*) orchards, and other agricultural lands [37,38]. The ecological health of the peninsula is also threatened by increasing pollution of its freshwater sources and shoreline [39]. The construction of large roads has caused a corresponding increase in pollutants that wash off roads during rain events and are carried into biologically rich streambeds and ecologically sensitive coastal waters [40,41]. Additionally, seawater around the peninsula is contaminated by toxins originating from abandoned mercury mines [33]. Polluted runoff and also erosion are growing problems because of the increase in number of severe rain events due to climate change. It is estimated that over 1,279,000 t/ha of the peninsula's soil was lost by erosion in 2010 [42].

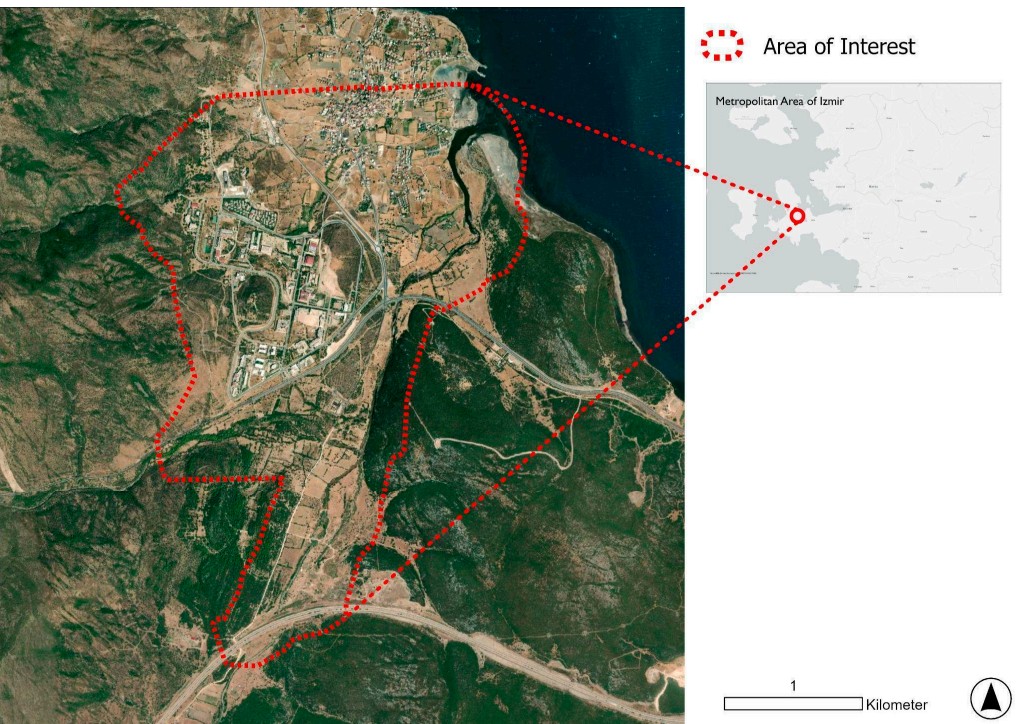

**Figure 1.** Area of Interest: IZTECH campus in the eastern Aegean region, on the Cesme-Karaburun Peninsula (Metropolitan Area of Izmir, Turkey). Source: Authors' elaboration based on ESRI ArcGIS Pro, licensed by Politecnico di Milano. Coordinate System WGS 84.

Our specific AOI is the campus of the Izmir Institute of Technology (IZTECH), a research institute and public university established in 1992 and located on approximately 150 hectares near the village of Gulbahce in the Izmir province. The built-up area of the campus is approximately 8.5 hectares, and it is constructed on the eastern slope of a rocky mountain covered with maquis [43]. The stratigraphy is dominated by a Miocene volcano-sedimentary succession, including several sedimentary and volcanic units developed on top of the basement rocks of the Karaburun Platform and Bornova Flysch Zone [44,45]. The average elevation of the built-up area of campus land is 88 m a.s.l., and the average slope in this area is approximately 10 degrees along the primary E-W pedestrian axis. After seasonal rains, many small streams flow through the central campus. In summer, the campus experiences high temperatures, drought, and wildfires. In winter, it is subject to strong winds and flooding. The region is also seismically active.

The indigenous vegetation of the campus is primarily evergreen sclerophyll shrubs, including prickly burnet (*Sarcopoterium spinosum*), thyme (*Thymus* spp.), oregano (*Origanum* spp.), lavender (*Lavandula stoechas*), bay laurel (*Laurus nobilis*), rock rose (*Cistus* spp.), kermes oak (*Quercus coccifera*), fig (*Ficus carica*), olive, and mastic (*Pistacia lentiscus*), with relatively small clusters of red pine (*Pinus brutia*), elm (*Ulmus minor*), and plane (*Platanus orientalis*) trees. During the construction of the campus, the topography was radically reshaped and large areas of the indigenous vegetation were greatly disrupted or removed. Wide areas of the surface were made impermeable through compaction of the soil during construction of roads, buildings, and walkways. In some locations, roads, pedestrian walkways, parking areas, and even large buildings were constructed over natural stream routes. In other locations, streams were channelized or attempts were made to divert the streams into underground pipes (Figure 2).

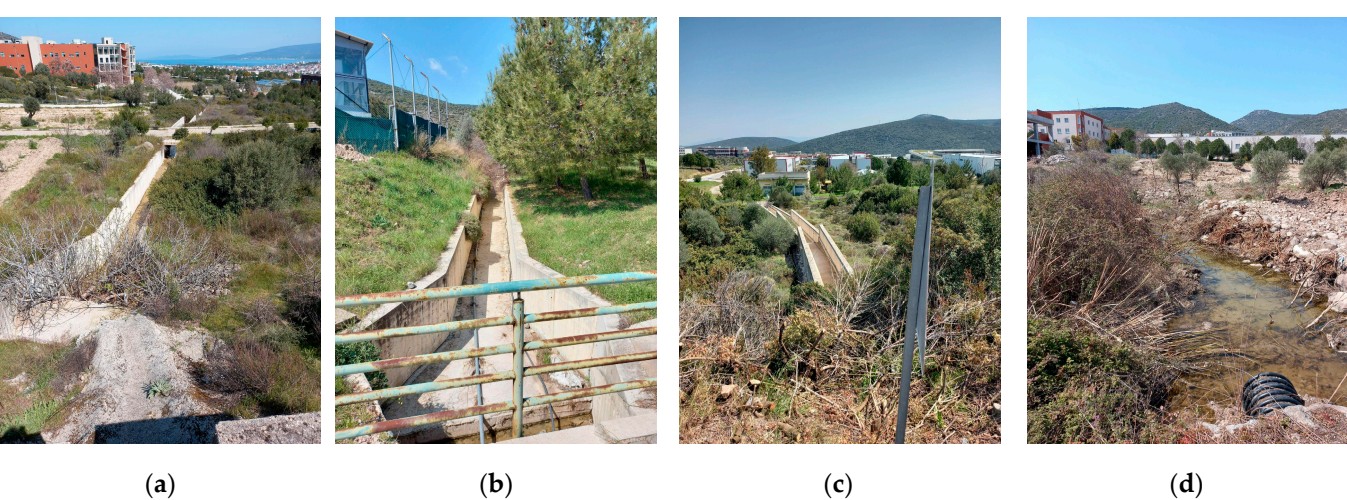

(**a**)　　　　　(**b**)　　　　　(**c**)　　　　　(**d**)

**Figure 2.** Channelization of streams in the campus: (**a**) main west–east channel, (**b**) main north–south channel, (**c**) interception point in the north campus fringe, (**d**) discharge point in the central campus. Source: author's pictures (S.S.).

Many personnel and students commute to the campus by bus. After arriving on campus, connections between buildings are difficult because of the steeply sloped terrain, a lack of climate-adapted outdoor spaces, and significant distances between buildings. These environmental challenges limit the spontaneous encounters of people on campus and, thus, negatively impact the campus community.

*1.2. Use of GIS-Based Ecological Analysis to Support Sustainability Initiatives*

GIS-based ecological analysis can be used as part of a Decision-Making Support Method to assess the environmental impacts of the settlement form, buildings, or operations of an area. As recently as 2017, two popular guideline and reporting frameworks for campus sustainability assessment, the Sustainability Tracking, Assessment, and Rating System (STARS) and the Global Reporting Initiative (GRI), lacked integration of GIS and the spatial dimension [11]. This research gap was addressed in a study by Alshuwaikhat et al., in which a GIS-based model for assessing the environmental sustainability of a university campus was proposed to evaluate the energy and water usage, solid waste management, transportation systems, and overall environmental quality of a campus [11]. GIS-based analysis was used to identify hot spots of high energy use in buildings, negative environmental impacts of various transportation systems, and estimate ratios of water usage and solid waste production for different locations on the campus. Results were used to produce a map of overall emissions levels and model future scenarios [11]. In a study of Manisa (Izmir/Turkey), Gulcin and Yilmaz (2020) demonstrated that using spatial data is highly effective for analyzing environmental changes. They employed a GIS-based model to quantify changes in ecological connectivity, and the output values

for land-use change indicated that most ecological connectivity losses occurred due to surface sealing [46]. They recommended that planning should be based on a detailed habitat map, that landscape analysis and assessment should include ecological connectivity, and that landscape metrics should be integrated into the spatial planning process. Ruiz et al. designed, constructed, and applied a GIS-based Spatial Decision Support System to land-use decisions and produced maps that revealed which zones were suitable for use as industrial areas, according to sustainability criteria [47].

It is not currently possible to empirically measure the processes of reduction in biodiversity or ecological connectivity, as this would be very time-intensive and would require long-term field measurements of vegetation and animal site specificity. However, quantifying imperviousness is often a good environmental proxy for the anthropogenic impact on the natural landscape, and quantifying imperviousness is much simpler than measuring the reduction in biodiversity or the reduction in ecological connectivity [48,49]. In our previous study of the IZTECH campus area, we used NDVI as a proxy for other soil degradation processes that occurred between 2021 and 2022. In that study, the maximum NDVI reduction (−32.9% to −29.4%) was recorded by surface sealing; compaction recorded a medium NDVI reduction (−24.3% to −26.6%), and erosion recorded the least reduction in NDVI (−14.5% to −17.2%) [43].

### 1.3. Nature-Based Solutions, and Using GIS-Based Maps to Measure Their Effectiveness

Nature-based Solutions are inherently economical; for this reason, they are suitable options for improving climate adaptation of large areas of land, especially in rural areas [50–52]. NbSs are multi-functional and can achieve a variety of ecological benefits at once [53,54]. For example, soil moisture can be increased by using permeable pavements to restore some of the absorptive function of soil as part of the natural water cycle, and an increase in soil moisture can contribute to biodiversity, fire resistance, wildlife connection, drought resistance, carbon sequestration, and reducing flooding [55–57]. A synthetic review by Marchioni and Bessiu indicated that the application of permeable pavements in urban areas is both practical and effective in reducing runoff volume and improving water quality [58]. The broad application of permeable pavements on the campus of Istanbul Technical University has resulted in improved stormwater management [59]. By using NbSs such as vine-covered trellises and green roofs, it is possible to reduce the heat-island effect. A reduced heat-island effect in hot climate zones can permit people to spend more time outdoors [60], which, in turn, may lead to greater social connection [61]. Baykal and Topal studied different platforms that deal with NbSs by employing an array of map layers, demonstrating the importance of using GIS-based thematic maps to determine the effectiveness of NbSs. They concluded that GIS-based thematic map layers can be effective tools for preparing future scenarios of NbSs [62]. In this study, a primary design goal was to improve habitat quality by revitalizing soil. We used GIS-based thematic maps in ecological analysis to predict the benefits of two NbSs applied to a masterplan design for a university in the eastern Mediterranean region. By using GIS-based thematic maps together with local values obtained from a high-resolution photogrammetric scan of the catchment, we were able to produce flow accumulation and habitat quality indexes and achieve a customized analysis of the Baseline Condition and two NbSs scenarios.

### 1.4. Goal of the Study

The primary goal of this study was to create a simple methodology for using modeling to predict the ecological benefits of specific NbSs that are applied in a particular location [57,63]. The study addresses the specific environmental challenges of a university campus, including its semi-arid climate, poor soil quality, seasonal flooding, habitat fragmentation, and loss of biodiversity [64,65]. In particular, this study aimed to identify economical methods of improving the absorption of rainwater into the ground and thus achieving a variety of ecological benefits, such as improved water quality, improved soil quality, and improved habitat quality. To improve the absorption of rainwater into the

ground, one common approach is to replace impervious surfaces such as packed earth or concrete with surfaces that can absorb water. This is known as "de-sealing". The proposed method of quantifying ecological indicators is a Decision-Making Support Method that can be used to help the campus achieve its aim of becoming a "sustainable green campus" [66].

## 2. Materials and Methods

Using Copernicus Services, we generated soil moisture (SM) index maps of the AOI for the years 2016 and 2022. We began with the year 2016 because that was the year of the first Copernicus acquisition. We compared the two SM maps to identify where natural green areas had been degraded due to construction activities. We then used scenario modeling to demonstrate that significant ecological benefits can be achieved with simple NbSs. For our first iteration of NbSs (Scenario 1, "Re-Classification of Land Use Land Ccover—LULC—Shrublands and Grasslands"), we added large plantations of groundcover composed of indigenous species of shrubs and grasses. We chose local plant species because of their capacity to increase soil moisture and revitalize the soil and also because native species are well-adapted to the climate and therefore require fewer inputs (e.g., water and fertilizer) than introduced vegetation. According to our first modeling results, we added a second NbS (Scenario 2, "Re-Classification of LULC—Shrublands and Grasslands, with De-Sealing") to improve the ecological benefits of Scenario 1.

Our working method was composed of the following steps (Figure 3):

1. Photogrammetric scanning of the AOI;
2. Downloading soil moisture maps from Copernicus for years 2016 and 2021;
3. Creating flow accumulation maps from (1) high-resolution Digital Surface Model (DSM) and (2) low-resolution Digital Elevation Model (DEM);
4. Mapping habitat quality (biodiversity), using auto-produced land-use–land-cover map and Copernicus images from 2016 and 2021;
5. Designing a preliminary NbSs masterplan;
6. Mapping habitat quality of the AOI with the first iteration of NbSs (Scenario 1), in which a relatively wide area was re-classified as "Groundcover, Shrubs and Trees";
7. Interpreting results and creating a performance-based iteration of NbSs (Scenario 2), in which de-sealing of select areas of existing pavement was added to Scenario 1 to reduce the edge effect of roads;
8. Comparing the results of analysis of the Baseline Condition (BC) with the first and second iterations of NbSs (Scenarios 1 and 2);
9. Revising the design strategy and creating a performance-based NbS masterplan, based on modeling results.

### 2.1. Data Acquisition and Processing

We commissioned a photogrammetric scan of the AOI with a resolution of 23 points per m$^2$. Data were acquired between 5 and 6 July 2021 using a DJI Phantom 4 Pro V2 drone and the following equipment: Spectra Precision ProMark 500 GNSS receiver and data collector with SurvCE software and a laptop equipped with Pix4D software. The drone-acquired data at an average distance from the ground surface of 185 m (5 cm ground sample distance). These data were used to generate a colored multispectral image (red, green, and blue bands) of the campus and its surroundings with a ground resolution of 30 cm, a LiDAR point-cloud dataset for tri-dimensional evaluation of the surfaces, including small vegetation, and a Digital Surface Model. The drone survey covered a catchment of 660 ha, which included the built-up area of campus, the eastern riverine and mostly plain areas, and, to the north, the deltaic system of the village of Gulbahce, which was settled between the rivers near the coast. Data from the scan were processed to produce raster and vector images.

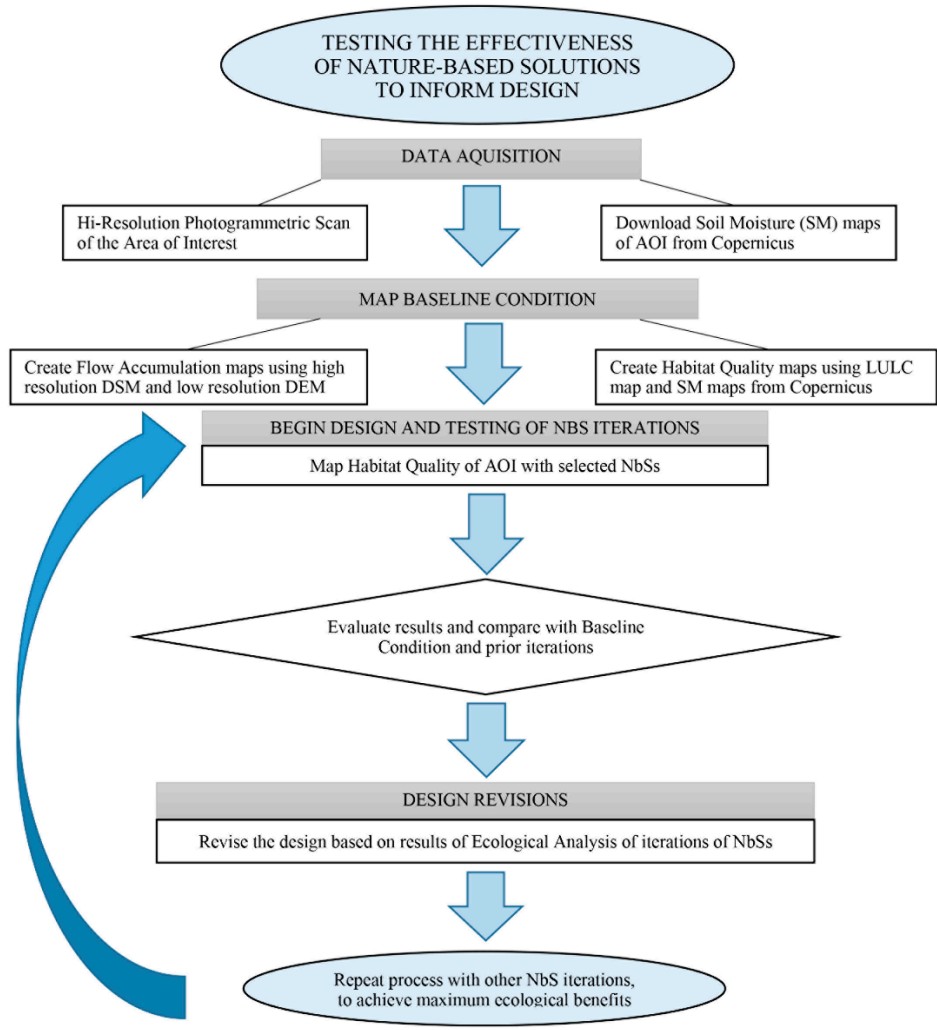

**Figure 3.** Workflow diagram. Source: author's elaboration (V.C.).

*2.2. Use of Copernicus Images*

Comparing the Copernicus images from 2016 and 2022, we observed changes that occurred to vegetation and soil during this period (Figure 4). It can be seen that a significant decrease in soil moisture occurred and that construction caused significant harm to the ecosystem through habitat fragmentation.

*2.3. Ecological Analysis of the Existing Study Area (Baseline Condition)*

2.3.1. Existent Land Use and Land Cover

The original photogrammetric data acquisition was post-processed to obtain a LULC characterization of the AOI. A supervised classification sampling was applied on a testing tile of the original acquisition to reduce the processing time [62–64]. Forty-five training points were used on the selected tile to set the algorithm of classification. After testing, this algorithm was applied to the entire acquisition to obtain a complete LULC map of the catchment. The final classification included six classes: soil (bare land), brush, grassland, trees, buildings, and roads. We used this simple classification as a modeling benchmark and set the final ground resolution of the LULC to 1 m. To avoid the edge effect (which is the bias created by modeling along the border of the catchment), we mosaicked the detailed LULC with a generic LULC, which was auto-produced by applying supervised classification sampling to a Copernicus tile of the western Izmir promontory (S2B_MSIL2A_20210328T085559) with a 10 m ground resolution. After this re-classification process, we merged the output with the more detailed LULC map obtained with the drone.

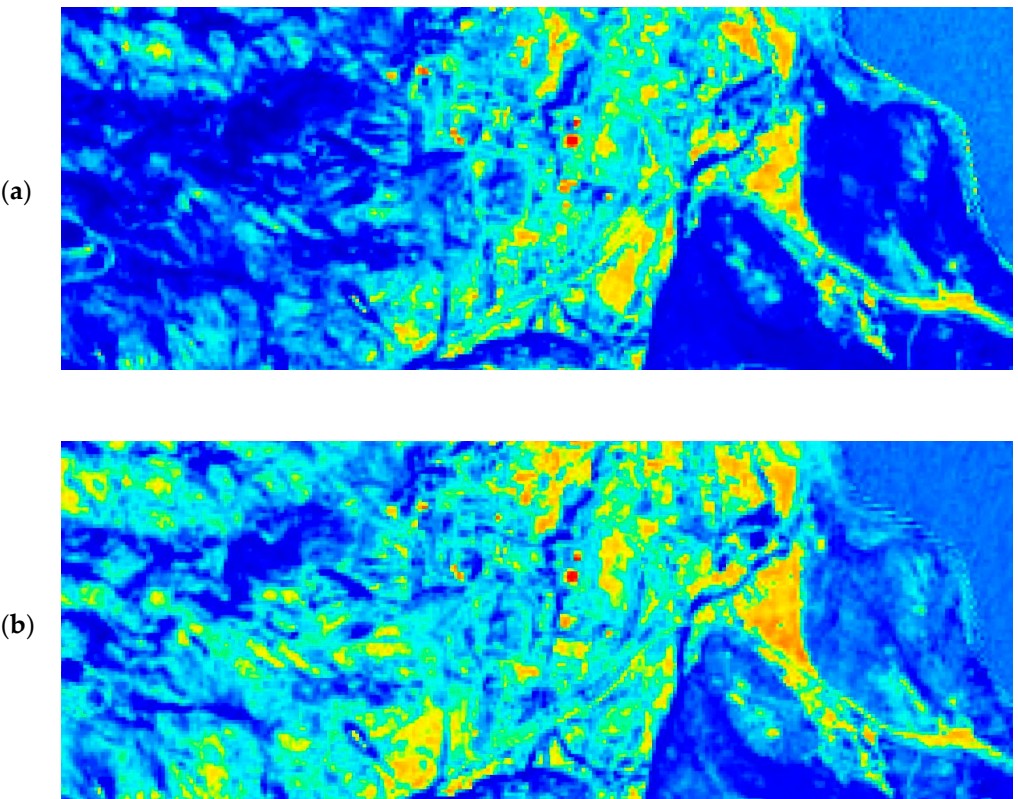

**Figure 4.** Soil moisture index map of the campus area (low moisture—dark red, high moisture—dark blue), (**a**) 15 July 2016 and (**b**) 12 July 2022. North-oriented. Scale of acquisition 1:2000. Source: Copernicus Sentinel data collection, acquired through Sentinel Hub services. Originally downloaded Copernicus Sentinel data processed by Sentinel Hub.

### 2.3.2. Modeling Habitat Quality of the Baseline Condition

Once the LULC was created, we employed InVEST (Integrated Valuation of Ecosystem Services and Tradeoffs) software [67] to model habitat quality (HQ) within the AOI as a proxy for supporting the ecosystem service-delivery capacity [57,68]. InVEST is a suite of models used to map ecosystem service-delivering capacity, observe how this can be subjected to flows, and evaluate the different benefits it can produce for people. The HQ model expresses the ability of an ecosystem to provide appropriate conditions for individual and population persistence [69]. The output reflects both the proximity of habitat to anthropic land uses and the threats caused by anthropic areas. The model can be used to obtain a map of suitable habitats for certain fauna, or it can be employed more generally to attain a general assessment of the quality of habitat for all species [70,71].

The HQ model requires (i) inputs on the LULC map, (ii) threats to habitats, articulated as the maximum distance over which each threat will affect a habitat, and (iii) habitat types and their unique sensitivities to threats. The InVEST HQ model uses habitat quality and habitat decay (DEG) as proxies to represent the biodiversity of a landscape. The HQ model combines maps of LULC with data on threats to habitats and habitat response. This combination of inputs generates a spatial map of HQ and DEG, which identifies areas where conservation/valorization/re-naturalization will most benefit natural systems and protect threatened areas. The final value of the HQ indicator ranges from 0 (no quality) to 1 (max quality); this value is relative to the catchment considered (the LULC extension). The digital input data (Input S1: habitat quality model) are attached in the Supplementary Materials section.

2.3.3. Flow Accumulation Analysis of the Baseline Condition

After modeling HQ, we used the Hydrology toolset of ArcGIS 10.8 (licensed by IZTECH) to evaluate water movements, based on the topography of the study area. Such analysis required main input data, which we gathered in three steps. The input data must represent the topographic status of a wide area, so our first step was to begin with a Digital Surface Model (DSM) with a scale of acquisition of 1:12,000. The average altitude offered by this DSM was approximately 121.6 m, with a maximum altitude of 212 m and a minimum altitude of 31.2 m. Digital models are pixel-based, so any gaps in the surface data can be understood as pits. This phenomenon is called a sink, and it can be misleading during analysis. The Fill tool allows such gaps in data to be filled and harmoniously merged with the surrounding terrain. We corrected our model using the Fill tool of the ArcGIS 10.8 toolset to resolve such small glitches.

After completing the corrections, our second step was to create a flow direction map that showed which directions water would flow into during rain events, based on the topography. Our third step was to use data from the flow direction map to perform a flow accumulation analysis and generate a raster map that showed where water would accumulate, based on the topography-related movement of water during rain events. We visually checked the results and deduced the movement characteristic of water in the area by visualizing a threshold flow accumulation set to 3000, based on our experience of water flows in the region on rainy days. We checked for reliability by comparing our results with a flow accumulation map that we created using a lower-resolution (20 m) DEM, which was produced by the European Union as part of the Copernicus Programme (Figure 5a). Flow accumulation has been preferred to a more detailed hydrological analysis because there has not been enough knowledge of the topsoil conditions in the catchment to understand exactly the biophysical dynamic of stormwater management. However, visualizing flow accumulation has been extremely useful in supporting the design process of the masterplan. In fact, there is no edge effect in the flow accumulation map that we created using the lower-resolution DEM, and this map provides an accurate representation of the spatial accumulation of stormwater flowing through the AOI. We produced a second flow accumulation map (Figure 5b) using a higher-resolution DEM (30 cm); we created this second map with data that we collected by photogrammetric scanning. The second map (Figure 5b) describes the waterways in much greater detail, but there is a significant edge effect because our photogrammetric scan included only the lower portion of watersheds that begin at much higher elevations to the west of the AOI.

*2.4. Creation of Preliminary NbS Masterplan*

After performing an ecological analysis of the Baseline Condition (Figure 6), we created a preliminary NbS masterplan for the built-up area of campus (Figure 7). We chose to work with NbSs because of their inherent low maintenance, sustainability, and economy, and because NbSscan perform multiple ecosystem services while harmoniously integrating with a natural setting [72,73]. We selected types of NbS that were appropriate for the local climate, topography, and soil conditions, i.e., semi-arid climate, sloped terrain, and clayey soil, with areas of packed or sealed surface. For these conditions, it is important to capture and absorb rainwater, reduce erosion, revitalize the soil, and reduce the heat-island effect.

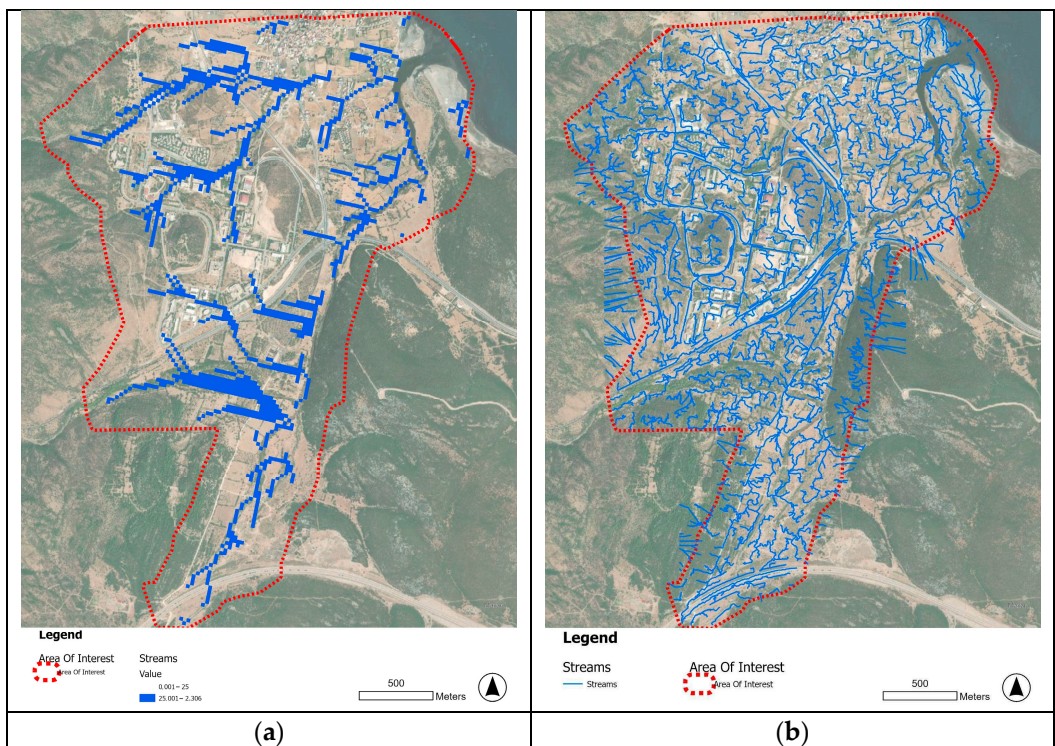

**Figure 5.** Flow accumulation output made using the Copernicus DEM at 20 m resolution (**a**) and flow accumulation output made using the drone survey at 30 cm resolution (**b**). Source: the Copernicus Sentinel data collection, acquired through Sentinel Hub services. Copernicus Sentinel data is downloaded and processed by users.

We began our Preliminary NbSs masterplan design by identifying dry areas within the campus, referring to the soil moisture map for 2021, which we downloaded from Sentinel Playground. For areas of campus with low soil moisture, we proposed new plantations of indigenous species to connect the existing natural areas. In our design, we included several types of NbSs that would increase stormwater absorption. We used our more detailed flow accumulation map (Figure 5b) to confirm the locations of water routes. Where desirable and possible, we redirected the existing water routes slightly to slow the flow, increase stormwater absorption, and reduce erosion. We designed continuous plantations of indigenous species along the banks of existing streams to create "green corridors" that would absorb stormwater, trap particulate matter, and reconnect fragmented areas of habitat.

We used drought-tolerant native species of shrubs and trees in xeriscape plantations to reduce the need for watering. For areas of bare soil and under trees, we indicated that indigenous species of grasses, succulents, and shrubs be used as groundcover to reduce erosion and improve soil moisture. This xeriscape groundcover would include local species such as terebinth (*Pistacea terebinthus*), oregano, rosemary (*Salvia rosmarinus*), sage (*Salvia officinalis*), rock rose, juniper (*Juniperus* spp.), laurel, thyme, and lavender. We designed xeriscape plantations of native trees along walkways to connect existing areas of shade. These tree plantations would include mastic, olive, red pine, kermes oak, elm, plane, and poplar (*Populus nigra*).

We designed compost areas near campus food service and landscaping operations for recycling waste vegetable materials and revitalizing the soil. Along existing pedestrian paths that receive excess sun in summer, we indicated lightweight, cable-stayed trellis structures covered with vines to create partial shade. These vines would include both evergreen and deciduous varieties, such as jasmine (*Jasminum officinale*), wisteria (*Wisteria sinensis*), bougainvillea (*Bougainvillea glabra*), passionflower (*Passiflora incarnata*),

honeysuckle (*Lonicera periclymenum*), trumpet vine (*Campsis radicans*), grape (*Vitis vinifera*), Virginia creeper (*Parthenocissus quinquefolia*), and climbing rose (*Rosa* spp.).

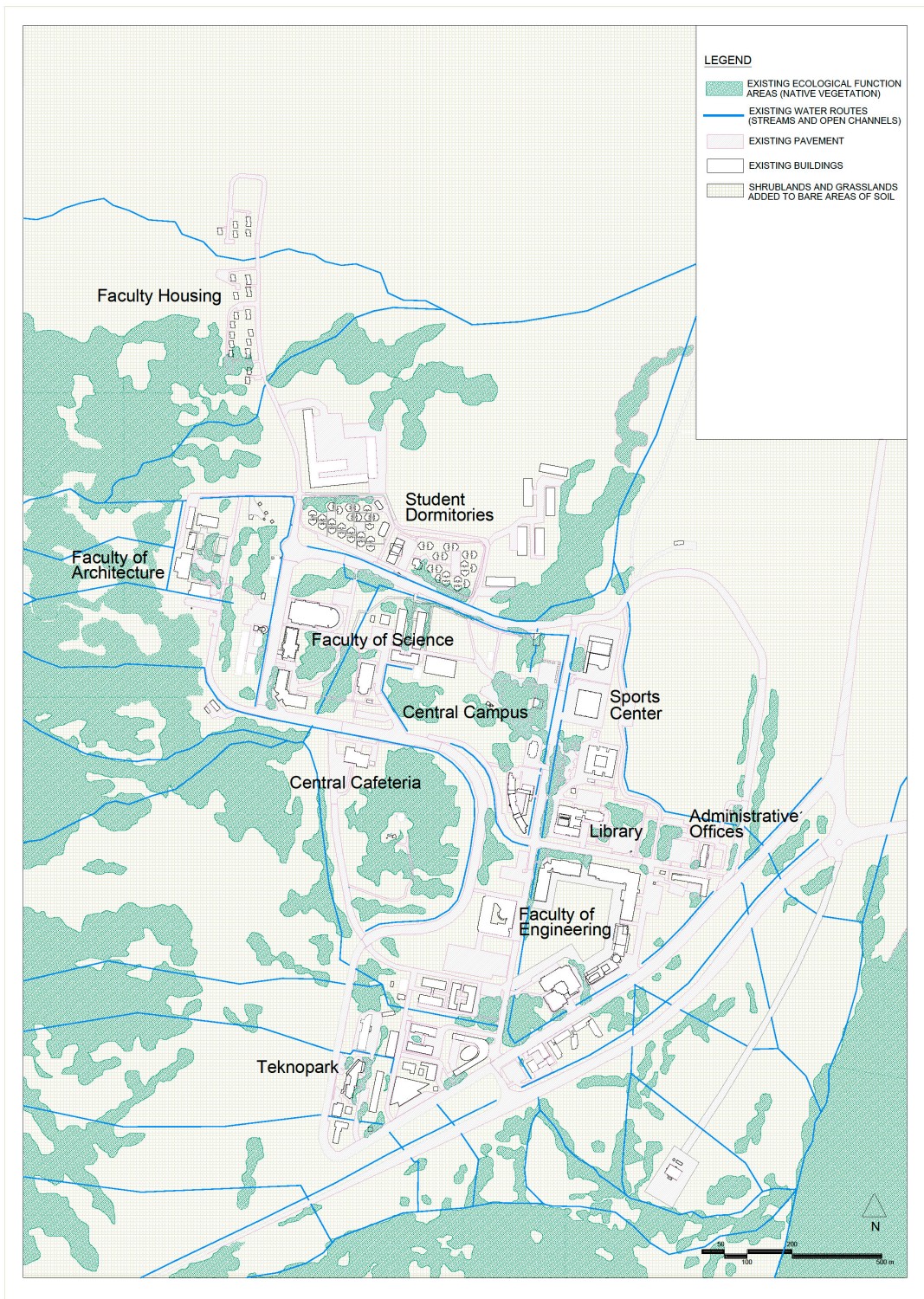

**Figure 6.** Plan of IZTECH campus, Baseline Condition (BC). Source: author (V.C.).

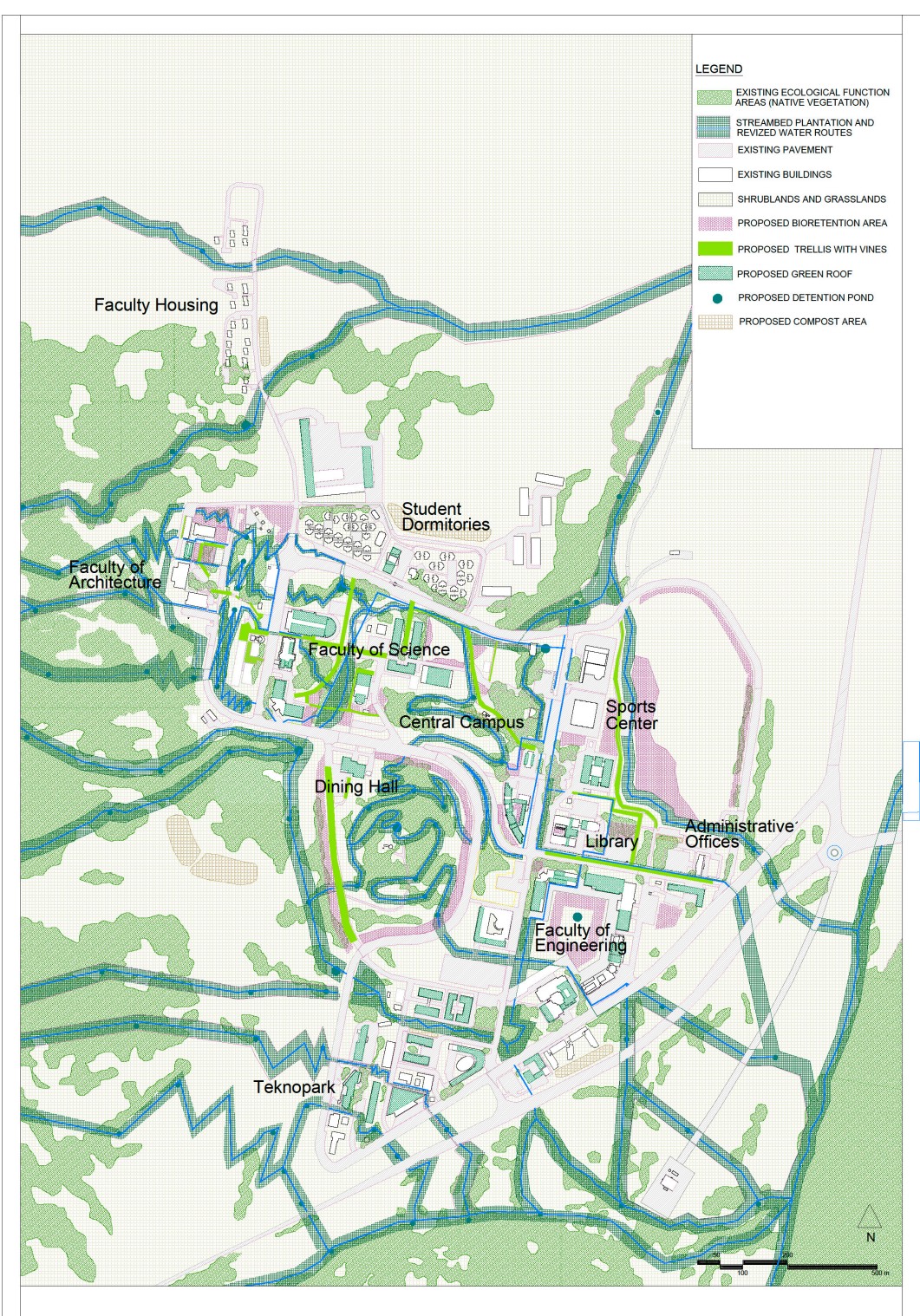

**Figure 7.** Preliminary NbSs masterplan. Source: author (V.C.).

We indicated zones along streams that would be enriched with xeriscape plantations of water-tolerant local species such as oleander (*Nerium oleander*), reed (*Phragmites australis*), elm, plane, poplar, willow (*Salix* spp.), and kermes oak. We recommended that the existing channelized stream beds should be re-naturalized to reduce flow peaks, improve water absorption, restore wildlife habitat, and connect fragmented natural areas.

The AOI is ideal for rain gardens because of its sloped topography. Increasing absorption allows soils to metabolize toxins in the runoff, thus improving water quality

downstream and preventing the discharge of the toxins into the sea. We designed bioretention areas to catch and absorb more rainwater; these would be rain gardens in open areas and bioswales along the edges of sidewalks and roads. The bioretention areas would be constructed with sloped sides and heavily planted with indigenous species to reduce runoff and filter pollutants. To pause the flow of stormwater, we designed detention ponds along the existing natural streams and in locations where water is directed to culverts in order to pass under the campus roads.

We designed green roofs for buildings on campus that have sufficiently large areas of open, flat roof to absorb more stormwater on-site and restore some of the biodiversity that was lost during the construction of the campus [74,75]. These green roofs would have a thick layer of soil to reduce the heat-island effect, and they would be planted with low-growth, drought-tolerant, indigenous species, such as oregano, juniper, lavender, thyme, and various succulents. Some species used in the green roof plantations should attract pollinators and provide habitat for local insects and birds.

## 2.5. Re-Classification and Ecological Analysis of Scenarios 1 and 2

To check whether the NbSs would increase HQ in the AOI, we modeled three different LULC configurations:

1. Baseline Condition (BC), derived from digitalization of the origins, per photogrammetric acquisition by drone;
2. Scenario 1 (S1), which included the first iteration of NbSs;
3. Scenario 2 (S2), which included the second iteration of NbSs and tested whether or not it was possible to reduce habitat decay along the main roads of campus by de-sealing in areas where roads caused a high degree of ecological interference with streams.

HQ modeling was used for two design iterations. Scenario 1 ("Re-Classification of LULC—Shrublands and Grasslands") proposed adding a relatively large area (over 100 ha) of indigenous groundcover vegetation. After interpreting the first habitat decay output, we suggested further improving the habitat condition by de-sealing a small amount of paved/impermeable land in certain areas where the anthropogenic impact on habitat was high (according to Scenario 1), specifically, in locations where natural streams had been interrupted by the construction of campus roads. Scenario 2 ("Re-Classification of LULC—Shrublands and Grasslands, with De-Sealing") proposed replacing the sealed road surface in these locations with permeable pavement. We then performed a second ecological analysis to determine the impact of this second NbS.

## 2.6. Preliminary NbSs Masterplan and Performance-Based NbSs Masterplan

We created the preliminary NbSs masterplan while modeling NbSs Scenarios 1 and 2. The modeling results show that a relatively limited area of de-sealing used in combination with a broad plantation of indigenous species of groundcover would provide potentially far greater ecological benefits than a broad plantation of indigenous groundcover alone. Informed by this result, we changed our design strategy for the performance-based NbSs masterplan (Figure 8). The performance-based NbSs masterplan includes many areas of de-sealing. We replaced areas of sealed surface with permeable pavement in locations where the campus roads cross natural streams, according to Scenario 2, and we designed the surfaces of many existing pedestrian walkways, car park areas, and ball courts as permeable pavements to significantly improve the ecological benefits. A more continuous flow of the campus streams would be possible if barriers that were constructed over streams were removed and the surface roads in these locations were replaced with simple bridges. Pedestrian walkways and bike paths could then pass freely under the roads alongside the streams in some locations to create continuous ecological and social connections throughout the campus.

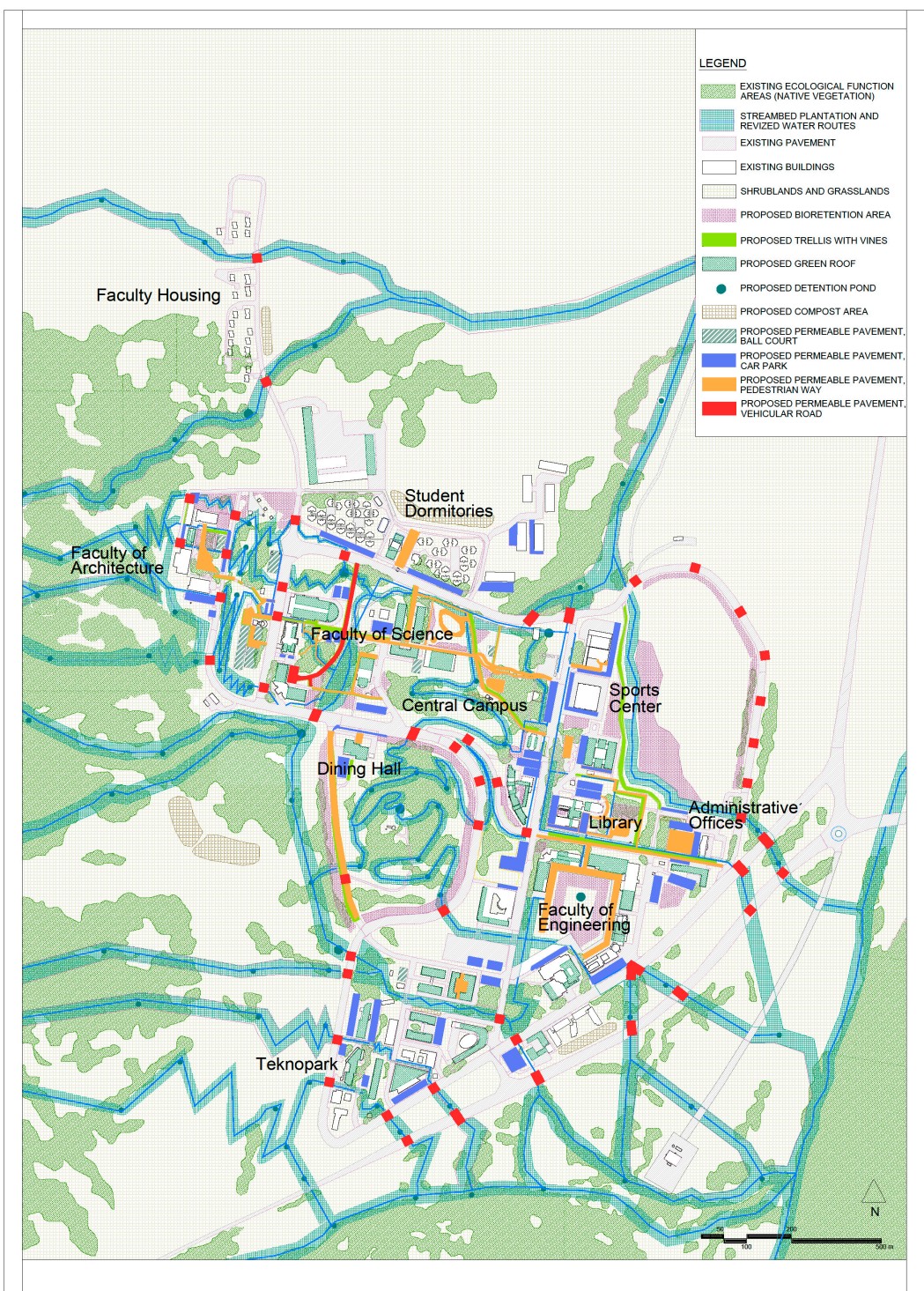

**Figure 8.** Performance-based NbSs masterplan. De-sealing is indicated in locations where roads pass over water routes and for some existing walkways, parking areas, and ball courts. Source: authors (V.C., N.S. and S.S.).

## 3. Results

*3.1. Baseline Condition (BC)*

3.1.1. LULC of Baseline Condition

The result of the supervised classification sampling for LULC classification based on the drone image acquisition generated the following statistics for the BC:

- 49.84% (329 ha) of the AOI is covered by bare land or soil without vegetation, including unpaved roads and compacted soil;
- 21.64% of the AOI is covered by dense vegetation (trees, 142 ha);
- 12.00% is covered by grasslands (79 ha);
- 10.31% is covered by shrubs and bushes (medium-density-vegetation land, 68 ha);
- 3.54% is covered by paved roads and parking areas (23 ha);
- 2.67% is covered by buildings (17 ha).

Construction of the campus has significantly altered the original morphology of the land while intercepting and impacting the natural flows of streams and small tributaries from the hill down to the main water basin; nevertheless, most of the AOI is still unbuilt. The built-up area of campus, therefore, represents a catchment with a great potential for biodiversity restoration.

### 3.1.2. Habitat Quality of Baseline Condition

The HQ model of the Baseline Condition displayed a minimum value of 0.12, a maximum value of 0.95, and an average value of 0.72, which is significantly higher than the mean value of 0.31 for HQ, which has been measured in the Izmir Urban Area [76]. Therefore, even though the campus land has been progressively urbanized, its average HQ index is more than double the value of that of the nearby dense urban catchment of the city of Izmir. This empirical consideration should be viewed as an important indicator of the great potential for conservation within the campus and the importance of conserving the existent values of the campus land. These values could be conserved by using afforestation and NbSs to connect and extend the green and biodiverse areas of campus. Areas of indigenous vegetation within the campus may be considered to be "ecological function areas" because they have high biodiversity values, according to our analysis.

### 3.2. Ecological Analysis of the First Iteration of NbSs Design (Scenario 1)

We analyzed both scenarios according to HQ indicators. Through this iterative process, we quickly achieved a significant projected increase in ecological benefits within the study area using NbSs. Scenario 1 would create a new 109 ha area of natural Mediterranean vegetation shrubs and bushes (medium-density-vegetation land) while diminishing by 16.51% the area of soil that has no vegetation and increasing the area of shrubs and bushes by 60.20%. Table 1 summarizes the differences seen in the habitat quality (HQ) and decay (DEG) indexes (mean change) between the actual configuration of the campus (BC) and Scenario 1.

**Table 1.** Summary of changes in the HQ and DEG indexes (mean change) from the BC to Scenarios 1 and 2. Source: author's elaboration (S.S.).

|  | Min | Max | Mean | Std Dev | NBC | Average Change from Baseline |
|---|---|---|---|---|---|---|
| HQ, Baseline | 0.120 | 0.950 | 0.720 | 0.170 | BC |  |
| HQ, Scenario 1 | 0.120 | 0.950 | 0.749 | 0.178 | S1 | 3.97% |
| HQ, Scenario 2 | 0.091 | 0.950 | 0.774 | 0.159 | S2 | 3.33% |
| DEG, Baseline | - | 0.147 | 0.012 | 0.027 | BC |  |
| DEG, Scenario 1 | - | 0.147 | 0.011 | 0.027 | S1 | −7.67% |
| DEG, Scenario 2 | - | 0.126 | 0.009 | 0.032 | S2 | −18.79% |

As seen in Table 1, application of the NbSs proposed in Scenario 1 ("Re-Classification of LULC—Shrublands and Grasslands") would guarantee an increase in HQ of about 4% to the campus area and a decrease in habitat decay of close to 8%. This means that new urban green areas on the campus can substantially mitigate the anthropogenic threats to natural elements (such as those natural features that are directly affected by the decay factors due to their proximity to infrastructure) while guaranteeing a higher HQ within the campus.

The modeling results demonstrate a second relevant result: a change in land use for 16.5% of the catchment can generate a 7.7% abatement of habitat decay. This means that if, in the future, a target of 10% abatement of habitat decay is set for the campus, then new green areas should be provided for at least 20% of the catchment. According to our model, the relative contribution of each new hectare of green area on the campus can guarantee an increase of 0.04% on the habitat quality index and a decrease of 0.07% on the Habitat Decay index.

*3.3. Ecological Analysis of the Second Iteration of NbSs Design (Scenario 2)*

After examining the first habitat decay output, we suggested a further improvement of the habitat condition by de-sealing the soil in areas where the anthropogenic impact of the roads on habitat was higher (Scenario 2—"Re-Classification of LULC—Shrublands and Grasslands, with De-Sealing"). We hypothesized that the impact would be higher in the areas along the campus roads where the natural streams had been interrupted by packed soil and sealed with impermeable pavement during road construction. For these locations, we re-classified 20 m diameter areas as "permeable pavement" and then performed a second ecological analysis to determine the potential impact of this NbS.

Scenario 2 created the same new plantations of indigenous vegetation as Scenario 1 and also reduced the anthropogenic footprint by de-sealing in these locations along the campus roads. Through this iterative process, we quickly achieved a significant projected increase in ecological benefits within the study area using NbSs. Table 1 summarizes the differences seen in the HQ and DEG indexes (mean change) from the Baseline Condition (HQ1, DEG1) to Scenarios 1 (HQ2, DEG2) and 2 (HQ3, DEG3).

In Table 1, it can be seen that Scenario 2 guarantees a 3.33% increase in the HQ index. This increase is only slightly less than that achieved by the BC with Scenario 1. What is surprising is how much the de-sealing reduced habitat decay. The second iteration of the model (Scenario 2) showed a decrease in habitat decay of more than 18%; this is more than double the benefit provided by the BC with Scenario 1. These results demonstrate that evaluating alternative NbSs design solutions through modeling can be useful in defining an overall strategy for reducing anthropogenic stress in a semi-natural catchment.

## 4. Discussion

Land degradation in the semi-arid Mediterranean region has greatly increased in the past 50 years due to a combination of natural phenomena, such as droughts and wildfires, and anthropogenic activity, such as deforestation, agriculture, and urbanization [77]. In a semi-arid climate such as the AOI, it is important to increase the absorption of rainwater and to revitalize compromised soil by improving its porosity. Soil moisture is closely related to biodiversity, and permeable soil can be expected to absorb more water, so it is imperative to increase the porosity of the soil to restore or increase biodiversity. In a semi-arid climate, improving soil porosity can be an effective strategy for recharging groundwater and slowing erosion. De-sealing the ground surface can improve water quality downstream by allowing runoff to be absorbed and toxins to be metabolized within the soil. Adobati and Garda enumerate the benefits of de-sealing in their study of the urban and semi-urban environment of Italy's Lombardy Region [78]. Maienza et al. demonstrated that soils quickly return to life after de-sealing, even without adding exogenous topsoil [79].

Where water is limited, concentrating new vegetation along existing natural water routes may be a sustainable strategy. Our preliminary design proposes the creation of self-sustaining "green corridors" throughout the AOI, which we created by planting indigenous species along existing stream routes, to reconnect fragmented natural areas, reduce erosion, and increase the absorption of rainwater. Such green corridors may connect the built-up area of the campus to the neighboring villages, with potentially important social benefits, in addition to their ecological benefits [80,81]. In some parts of the AOI, the soils are clayey and difficult to infiltrate; in these locations, the NbSs should use engineered soils with sand or organic material added to absorb water more readily.

Landscape design should be informed by knowledge about both the broader ecosystem and specific local conditions. By combining lower-resolution data that we downloaded from Copernicus with higher-resolution data from a photogrammetric scan, we were able to benefit from both the broader view provided by the Copernicus images and the greater sensitivity of the scan. At the start of the design of our masterplan, we consulted a soil moisture map produced from Copernicus data, and we used it to identify intact areas of indigenous vegetation and pinpoint areas of bare and compacted soil within the AOI. We based our preliminary masterplan design on reconnecting fragmented natural areas, protecting and enhancing the ecological function of the natural streams, and increasing soil moisture.

We demonstrated that it is possible to quickly perform iterations and compare the impacts of different NbS options for a specific location. In Scenario 1 ("Re-Classification of LULC—Shrublands and Grasslands"), we tested the impact of adding large plantations of indigenous grasslands and shrublands. In Scenario 2 ("Re-Classification of LULC—Shrublands and Grasslands, with De-Sealing"), we added the de-sealing of select areas of existing impermeable pavement. Ecological analysis demonstrates that various benefits can be expected from these two NbSs, including increased soil moisture, reduced habitat decay, and increased biodiversity.

Our models indicate that plantation of indigenous species of groundcover vegetation combined with a reduction in anthropogenic footprint in strategic locations (Scenario 2) yields much greater ecological benefits than plantation of indigenous species of ground-cover alone, even when the added vegetation is planted over a large area. The significant increase in ecological benefits provided by Scenario 2 indicates that additional de-sealing may be a highly effective strategy for achieving substantial ecological benefits for the campus. In our performance-based NbSs masterplan, the existing ball courts, pedestrian walkways, and parking areas were also designed with permeable pavement. The ecological impact of each of these additional changes should be studied.

Benefits that could be expected from the NbS modeled in Scenario 1 include an increase in soil moisture and a reduction in habitat decay. Additional benefits might include increased biodiversity, increased resilience to drought and fire, improved stormwater absorption and erosion control, and reduced heat-island effect. These primary benefits could generate secondary benefits, such as an increase in time spent outdoors on campus, which would contribute to social connectivity. Our models were highly simplified, yet we were able to observe how the proposed NbSs would impact the ecological indicators of habitat decay and flow accumulation. The modeling indicates that a broad plantation of indigenous species of groundcover vegetation combined with a reduction in anthropogenic footprint (Scenario 2) yields much greater ecological benefits than a broad plantation of indigenous species of groundcover alone. The first results suggest that using de-sealing to further reduce the anthropogenic footprint would provide significant ecological benefits in terms of improved habitat quality and reduced habitat decay.

### 4.1. Limits and Potentialities

This work sought to promote the utilization of detailed digital data to support an adaptive and sustainable design process for campus transformation. Our ecosystem modeling session used a simplified LULC categorization, but the geometrical precision of the data we acquired by photogrammetric scanning was an asset to understand precisely where NbSs were needed.

Our simulated modeling Scenario 1 simply reduced the amount of bare land and demonstrated how HQ and DEG changed accordingly. Modeling always represents a simplification of reality, and we enormously reduced the quantity of existent micro-land-use typologies that characterize the catchment. For example, our simple categorization of "shrublands and grasslands" included uncultivated land in the neighboring village and vacant plots within the constructed part of the campus. Moreover, our categorization of

"bare land" included rocky formations scattered across the hill in the unconstructed area of campus plus land compacted by construction vehicles within the constructed area.

Our simulated modeling Scenario 2 reduced the anthropogenic footprint in specific locations where the stream routes had been interrupted by construction of primary campus roads. In these locations, the negative anthropogenic impact was in part due to sealing of the surface; therefore, we proposed that permeable pavements be used to replace existing impermeable pavements in these areas.

Instead of using modeling to obtain an objectively valid measure of biodiversity on campus (which, in any case, is very difficult to model), we preferred to observe the changes between the designed alternatives. Therefore, our work should be considered a Decision-Making Support Method that can guide the design process by quickly looking at iterations of a model during design. Our simplified LULC model allowed us to save computational modeling time and quickly produce ready-to-use biophysical models that aided decision making. We emphasized modeling iterations instead of absolute modeling reliability.

*4.2. Future Studies*

To eliminate edge effect and achieve a more precise flow accumulation analysis of the AOI, it is necessary to gather more precise data on the rest of the campus watershed, which extends westward a horizontal distance of approximately 200 m to a peak elevation roughly 300 m higher than the western edge of our study area. It is likely that using more precise data on the entire watershed would alter outcomes of the modeling.

Each one of the NbSs shown in our performance-based NbSs masterplan could be modeled and the results interpreted to discern its ecological benefits. Using the proposed method, it would be possible to model the effects of redirecting the streams or re-naturalizing the channelized streams in order to observe the effects of these NbSs on soil moisture and habitat quality. It would be revealing to model further reductions to the anthropogenic footprint, such as those that we have indicated in the performance-based NbSs masterplan, specifically, changing existing pavements to permeable pavements in car park areas, ball courts, and pedestrian paths. These further studies would provide a better understanding of the ecological impacts of de-sealing. The results of further modeling and analysis could be used to support the development of a long-term strategy of gradually replacing all impermeable pavements on campus with permeable pavements, re-naturalizing the campus streams, and revitalizing the campus soil. Each of these implementations would contribute to the university's resilience against negative effects of climate change. Other follow-up studies might include an analysis of the embodied energy and actual cost of the proposed NbSs. Such analysis would be useful, e.g., in comparing NbSs with contemporary "industry standard" methods of constructing stormwater channels and controlling erosion.

Further studies could also include modeling the impacts of the proposed NbSs over different periods of time, after plantations reach maturity and secondary growth occurs. Other specific ecological impacts of individual NbSs could be analyzed, such as drought resistance, resilience to wildfire, erosion control, and reduction in the heat-island effect. The precision of the models could be improved by using data collected from a detailed survey of indigenous plants in the study area.

**5. Conclusions**

In this paper, we demonstrate the potential of using GIS-based ecological analysis tools together with detailed data about a specific location to quantify the expected ecological impacts of NbSs and thereby support decisions about land-use policy and environmental design. We modeled the ecological impact of two simple Nature-based Solutions for a semi-rural university campus in the eastern Mediterranean region, which is highly vulnerable to climate change and suffers many negative effects such as water scarcity, wildfires, and loss of biodiversity. We used larger-scale data collected from satellite images together with finer-scale data collected by photogrammetric scanning to reveal two indicators of ecological health: soil moisture and HQ. We performed ecological analysis to test the impact

of two NbSs (Scenarios 1 and 2). Our models demonstrated that the proposed NbSs would provide significant ecological benefits, including increased soil moisture, increased HQ, and decreased DEG. In addition, our modeling results indicate that the primary causes of habitat decay in the AOI are anthropogenic features such as roads, other pavements, packed earth, and buildings.

Quantification of the ecological benefits of NbSs can provide decision support for designers and policymakers working towards sustainability goals. Using the proposed method, such decisions can be based on precise information about expected changes in ecological indicators such as soil moisture and habitat quality. Our method of analysis produces clear comparisons that can be used to increase public support for environmental design and land-use policy decisions. First, modeling results quickly demonstrated that de-sealing combined with the plantation of indigenous shrublands and grasslands provided much greater ecological benefits than such plantation alone. Such early results are valuable for informing an NbSs design strategy that seeks to achieve the maximum possible ecological benefits for a specific location.

**Supplementary Materials:** The following supporting information can be downloaded at https://www.mdpi.com/article/10.3390/cli11060116/s1. Input S1: Habitat Quality model.

**Author Contributions:** Conceptualization, V.T.C., S.S., N.S. and A.F.; Data curation, S.S.; Formal analysis, S.S. and B.A.; Funding acquisition, V.T.C., S.S., N.S. and A.F.; Methodology, V.T.C. and S.S.; Validation, S.S.; Visualization, V.T.C. and N.S.; Writing—original draft, V.T.C. and S.S.; Writing—review and editing, V.T.C., S.S., N.S. and A.F. All authors have read and agreed to the published version of the manuscript.

**Funding:** This research was funded by an IZTECH 2021 Science Research Project (Bilism Arastirma Projesi) grant, no. mimfak-bap-0088, project name "IYTE Living Laboratory and Ecological Park".

**Data Availability Statement:** The authors used the Copernicus Sentinel data collection, acquired through Sentinel Hub services.

**Acknowledgments:** Drone scanning and post-processing of data was performed by Raşıt Torun. A partial survey of plants in the AOI was performed by students from the Department of Molecular Biology and Genetics, Faculty of Science, IZTECH.

**Conflicts of Interest:** The authors declare no conflict of interest. The funders had no role in the design of the study; in the collection, analyses, or interpretation of data; in the writing of the manuscript; or in the decision to publish the results.

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
