# Peer review of "De-Sealing Reverses Habitat Decay More Than Increasing Groundcover Vegetation"

_climate, doi:10.3390/cli11060116_

Round 1

Reviewer 1 Report

The work deals with a topic of great interest in a complete and exhaustive way, well defining the complexity of the topics involved. The work can make an important contribution to the state of the art in the field of assessments of the state of habitats and of the hypothetical scenarios from a design point of view. The identified methods allow rapid processing and savings in terms of resources, including computational ones.

In the introduction, the objectives are clear for the reader and the sources cited are updated.

The methodology can be provide a valid tool to the subjects to whom it is addressed and is coherent, as well as well explained. In particular, public administrations and their respective consultants can find a valid evaluation tool in the methodology reported.

The discussion is well supported by the results they have obtained and is very articulated. The goals that were set have been achieved.

The conclusions are operational and practical with respect to the objectives, and also indicate the potential evolution of the tools and method.

In my opinion the work can be published immediately, I report only a few minor revisions.

Minor revisions

Line 211 – develop the acronym “LULC” out in full, and then delete it from line 262-263.

Reviewer 2 Report

The article could be accepted with minor reviews. It shows good scientific soundness, and it is significant for both research and planning practices. Moreover, it is very clear and well written.

However, there are a few aspects and details that could be improved, Find them in the following list, along with their reference line:

Line 16 – Consider adding “and mitigation” after adaptation, as the dual role of NbSs has been recognized in the scientific literature.

Line 18, 22, 53, … , 226 and further – More attention to capital letters should be paid, e.g. “applied to a university masterplan”, two NBSs scenarios, … , preliminary NbSs, …

Line 25 – A definition or an explanation for “anthropogenic footprint” should be provided, or, as alternatives, 1) a more specific term or 2) some examples could be added.

Line 25 – Consider writing Habitat Decay in lowercase when not associated with “index”, “indicator” or similar.

Line 64-65-66 – It should be clarified that you mean that the average temperature between the years 1938-2018 was 17.9 °C, and not that the average temperature was the same for all those years. Therefore, and not “but”, the average temperatures demonstrate a trend of increasing.

Line 67 – Consider adding a brief explanation about the main characteristics of these types of vegetation.

Line 128-129 and further (e.g., 360-361, 362-363, 368-369, 371-372) - Consider adding the scientific names of the plants.

Line 143 – “:” instead of “.”, or “main” should be capitalized.

Lines160-162 – Year or bracket reference missing for “Glucin and Yilmaz”.

Line 171-172 – Consider being more precise and adding a reference.

Line 183 – Functions.

Line 185 – The link between soil moisture and flooding link might be unprecise. An increase in soil moisture is a consequence of a higher permeability, i.e., higher (rain)water infiltration and storage capacity of the soil.

Line 194 – A clearer explanation of what is mean by “thematic map layers” in this context could be useful.

Line 211 – Consider providing the meaning of the acronym LULC when it is first mentioned, i.e., here, and not at line 224.

Line 224 – You provide an explanation for “Habitat Quality”, i.e., biodiversity. Considering moving the explanation when you first mention it.

Line 229 – Interpretating à Interpreting

Figure 3 – Missing “s” in “NbS”, in the lowest circle.

Figure 4 – Missing north, scale and reference with respect to the whole area.

Line 270 – A specification about the edge effect might be useful.

Line 312-314 – Did you take into consideration the idea of performing a hydrological analysis of the area with reference to the existing conduits and to real precipitation data, instead of performing only a Flow Accumulation analysis? Consider explaining the reasoning behind your choices.

Line 386 – Consider adding a reference concerning the role of green roofs in mitigating the UHI effect.

Line 405 – Please provide a clarification of what you mean by a “small amount of de-sealing”. A general definition of the term “de-sealing” itself might be useful as well.

Line 408-410 – A specification about the proposed de-sealing interventions should be provided. Do you foresee only a replacement of the top layer of the soil, or have you considered a deeper restoration of soil as well?

Line 470 – You introduced two acronyms for “Habitat Quality and Decay”. Consider introducing them sooner and always using them afterwards.

Line 477 and 499 – A specification about the edge effect might be useful here as well.

Line 561 – Missing “s” in NbSs.

Reviewer 3 Report

Very interesting, applied work using multiple approaches.

Since InVEST model was the main software tool that was used for modeling, I would expect more elaboration on how the integrated methods/tools were used.  For example, the Habitat Quality index as correctly indicated requires threats to habitats. These treats should be integrated into the model with an appropriate treats table (type of treat, max distance, weight etc.). 

 So, I think that the authors should provide more details on how the calculations were done and publish all the additional parameters they used. If the overall approach/purpose is to combine qualitative and quantitative data analysis, more info should be given on the methods used for the model. Maybe create an appendix with the additional info in case they can not be added to the main text.

 Additionally, the suggested citation for InVEST  is “Natural Capital Project, 2022. InVEST 3.13.0.post6+ug.g6b07b42 User’s Guide. Stanford University, University of Minnesota, Chinese Academy of Sciences, The Nature Conservancy, World Wildlife Fund, and Stockholm Resilience Centre.”, as referred in their official webpage.

Reviewer 4 Report

Modeling and assessing ecosystem sustainability using GIS-based approach is a growing potential field in scientific research, especially in these locations countering causing ecological threats such as droughts or wildfires. The authors used ecological analysis to predict the benefits of two NbSs applied to a university Masterplan, and got the results indicate that anthropogenic features are the primary cause of Habitat Decay and that decreasing anthropogenic footprint reduces Habitat Decay significantly more than adding vegetation.

The comments are as follows.

(1)    There are many language errors in the paper. It is highly recommended to find a native speaker to revise the manuscript.

(2)    In the Introduction, more detailed methods related to the research topics of this paper should be added and addressed.

(3)    Experimental results should be addressed using quantitative metrics. For example, the significant decrease in soil moisture in Figure 4.

There are many language errors in the paper. It is highly recommended to find a native speaker to revise the manuscript.
